# Topical Treatments and Their Molecular/Cellular Mechanisms in Patients with Peripheral Neuropathic Pain—Narrative Review

**DOI:** 10.3390/pharmaceutics13040450

**Published:** 2021-03-26

**Authors:** Magdalena Kocot-Kępska, Renata Zajączkowska, Joanna Mika, David J. Kopsky, Jerzy Wordliczek, Jan Dobrogowski, Anna Przeklasa-Muszyńska

**Affiliations:** 1Department of Pain Research and Treatment, Jagiellonian University Medical College, 31-008 Krakow, Poland; jan.dobrogowski@uj.edu.pl; 2Department of Interdisciplinary Intensive Care, Jagiellonian University Medical College, 31-008 Krakow, Poland; renata.zajaczkowska@uj.edu.pl (R.Z.); j.wordliczek@uj.edu.pl (J.W.); 3Department of Pain Pharmacology, Maj Institute of Pharmacology, Polish Academy of Sciences, 31-343 Krakow, Poland; joamika@if-pan.krakow.pl; 4Institute for Neuropathic Pain, 1056 SN Amsterdam, The Netherlands; Info@neuropathie.nu; 5Department of Anesthesiology, Amsterdam University Medical Center, 1081 HV Amsterdam, The Netherlands

**Keywords:** localized neuropathic pain, ion channels, receptors, peripheral mechanisms, topical treatment, peripheral sensitization, lidocaine, capsaicin, BTX-A, polyneuropathy, peripheral neuropathic pain

## Abstract

Neuropathic pain in humans results from an injury or disease of the somatosensory nervous system at the peripheral or central level. Despite the considerable progress in pain management methods made to date, peripheral neuropathic pain significantly impacts patients’ quality of life, as pharmacological and non-pharmacological methods often fail or induce side effects. Topical treatments are gaining popularity in the management of peripheral neuropathic pain, due to excellent safety profiles and preferences. Moreover, topical treatments applied locally may target the underlying mechanisms of peripheral sensitization and pain. Recent studies showed that peripheral sensitization results from interactions between neuronal and non-neuronal cells, with numerous signaling molecules and molecular/cellular targets involved. This narrative review discusses the molecular/cellular mechanisms of drugs available in topical formulations utilized in clinical practice and their effectiveness in clinical studies in patients with peripheral neuropathic pain. We searched PubMed for papers published from 1 January 1995 to 30 November 2020. The key search phrases for identifying potentially relevant articles were “topical AND pain”, “topical AND neuropathic”, “topical AND treatment”, “topical AND mechanism”, “peripheral neuropathic”, and “mechanism”. The result of our search was 23 randomized controlled trials (RCT), 9 open-label studies, 16 retrospective studies, 20 case (series) reports, 8 systematic reviews, 66 narrative reviews, and 140 experimental studies. The data from preclinical studies revealed that active compounds of topical treatments exert multiple mechanisms of action, directly or indirectly modulating ion channels, receptors, proteins, and enzymes expressed by neuronal and non-neuronal cells, and thus contributing to antinociception. However, which mechanisms and the extent to which the mechanisms contribute to pain relief observed in humans remain unclear. The evidence from RCTs and reviews supports 5% lidocaine patches, 8% capsaicin patches, and botulinum toxin A injections as effective treatments in patients with peripheral neuropathic pain. In turn, single RCTs support evidence of doxepin, funapide, diclofenac, baclofen, clonidine, loperamide, and cannabidiol in neuropathic pain states. Topical administration of phenytoin, ambroxol, and prazosin is supported by observational clinical studies. For topical amitriptyline, menthol, and gabapentin, evidence comes from case reports and case series. For topical ketamine and baclofen, data supporting their effectiveness are provided by both single RCTs and case series. The discussed data from clinical studies and observations support the usefulness of topical treatments in neuropathic pain management. This review may help clinicians in making decisions regarding whether and which topical treatment may be a beneficial option, particularly in frail patients not tolerating systemic pharmacotherapy.

## 1. Introduction

Neuropathic pain (NP) in humans arises as a consequence of a lesion or disease of the somatosensory nervous system [1] and affects 7–10% of the world population [2]. Patients suffering from chronic NP are characterized by higher health care utilization, higher risk of comorbidities such as depression, anxiety, and sleep disturbances, and lower quality of life compared to patients with chronic non-neuropathic pain [3]. The negative impact of NP on patients’ functioning and quality of life results, among others, from an unsatisfactory analgesic effect of neuropathic pain treatments. In clinical practice, most of the recommended pharmacological agents have moderate efficacy with a high number needed to treat (NNT) [4]. The NNT is the number of patients that need to be treated compared to placebo in a clinical trial to achieve, in one patient, at least 50% pain relief. The recommended oral pharmacotherapy is associated with a high risk of drug–drug interactions and side effects, potentially interfering with the analgesic effect, and limiting patients’ satisfaction. Non-pharmacological methods may be either little or moderately effective, or not available in clinical practice. Therefore, despite recent progress in developing new treatments of NP, many patients remain refractory to, or intolerant of, existing pharmacological and non-pharmacological therapies [5,6]. New drugs and/or routes of administration improving the effectiveness of NP management are constantly being sought. Nowadays, topical routes of analgesics administration are gaining popularity in pain medicine since topical treatments have an excellent safety profile and preference compared to systemic drugs in patients with peripheral NP [5,6]. Thus, in terms of safety, most vulnerable groups of subjects suffering from NP (e.g., elderly, frail patients) may particularly benefit from topical, rather than systemic, treatments [7].

This literature review aims to present the evidence from clinical and preclinical studies on possible molecular/cellular mechanisms of various topical treatments utilized in patients with peripheral NP, their modes of action in the peripheral neuronal and non-neuronal cells, and their clinical effectiveness. The current knowledge on topical treatments, possible molecular mechanisms, and their effectiveness in clinical practice is crucial for health care professionals dealing with patients suffering from peripheral NP.

We searched PubMed for papers published from 1 January 1995 to 30 November 2020. The key search phrases for identifying potentially relevant articles were “topical AND pain”, “topical AND neuropathic”, “topical AND treatment”, “topical AND mechanism”, “peripheral neuropathic”, and “mechanism”. We extracted all clinical trials, retrospective studies, case reports, narrative reviews, systematic reviews, and experimental studies in vivo and in vitro; double reports were left out. The result of our search was 23 randomized controlled trials (RCT), 9 open-label studies, 16 retrospective studies, 20 case (series) reports, 8 systematic reviews, 66 narrative reviews, and 140 experimental studies. Our review did not include studies indexed in databases other than PubMed.

## 2. Topical Treatments in Patients with Neuropathic Pain

The somatosensory nervous system may become injured at the peripheral or central level, which can result in peripheral or central NP, respectively. Common conditions associated with peripheral NP in humans include postherpetic neuralgia (PHN), painful diabetic neuropathy (PDN), trigeminal neuralgia, painful radiculopathy, HIV-associated neuropathy (human immunodeficiency virus), post-amputation pain, chemotherapy-induced peripheral neuropathy (CIPN), and peripheral nerve injury pain such as carpal tunnel syndrome or postsurgical NP. Stroke, spinal cord injury, and multiple sclerosis may, in turn, result in central NP syndromes [8].

The peripheral somatosensory nervous system may be damaged at several levels—peripheral nerve endings, axons, or cell bodies in the dorsal root ganglions (DRG)—in multiple ways, such as mechanically, thermally, chemically, and through infectious factors. In humans, the exact mechanisms of NP generation are not fully elucidated. However, preclinical studies with NP animal models gave some insight into the peripheral mechanisms involved [9]. The injury of peripheral neurons, independently of the cause and level of the damage, induces complex functional and structural changes not only in neurons (sensory and motor) and glial cells but also in non-neuronal cells such as keratinocytes and immunocompetent cells (e.g., macrophages, mast cells, neutrophils). The complex neuro–immune–cutaneous interactions, their changes observed after peripheral nerve injury, and their role in generation and maintenance of NP were reviewed in our paper published recently [10].

Although only a few molecular/cellular mechanisms of NP in humans are directly elucidated [10], the following preclinical and clinical findings justify the utilization of topical analgesics in clinical practice.

Input from hyperexcitable peripheral neurons is crucial for development, modulation, and maintenance of NP [11].Upon physiological nociception, peripheral neurons exert complex interactions with immunocompetent cells and keratinocytes via neuropeptides, neurotransmitters, cytokines, and other signaling molecules acting on corresponding ion channels or receptors (Figure 1) [12]. Once pathological conditions (i.e., nerve injury, inflammation) occur, these interactions result in overactivation and disturbed functioning of neuronal and non-neuronal cells, finally contributing to neuronal hyperexcitability, peripheral sensitization, and pain [11].Inhibition of peripheral sensitization can diminish and/or abolish the signs and symptoms of central sensitization in humans with peripheral and central NP [13,14].

A peripheral lesion of the nervous system induces more localized signs and symptoms of NP (e.g., allodynia, hyperalgesia) compared to a central lesion, which might be targeted by analgesics applied topically [15]. To improve daily clinical practice and to identify patients for whom a topical treatment should be considered, the definition of “localized neuropathic pain” (LNP) has been proposed [16]. LNP is a type of NP that is characterized by a consistent and circumscribed area(s) of maximum pain, associated with negative or positive sensory signs and/or spontaneous symptoms characteristic of neuropathic pain [16] and is felt superficially [15]. The circumscribed area of LNP is not larger than a letter-size piece of paper and may be diagnosed in 60% of NP patients, and up to more than 80% of PHN patients [16]. However, epidemiological data are scarce and estimation of LNP prevalence is broad [2]. Nevertheless, the proposed definition of LNP may help clinicians to optimally select the patients being best candidates for topical treatments.

Clinical data reviewed by Finnerup et al. [5] and Moisset et al. [6] show relatively good evidence supporting the use of topical analgesics in patients with LNP, especially in view of the low risk of side effects associated with this route of administration. Based on the current literature, the best evidence coming from randomized clinical trials (RCTs) to treat LNP with topical analgesics is for 5% lidocaine patches, 8% capsaicin patches, and botulinum toxin A (BTX-A) intradermal injections [5,6]. A recent review and clinical recommendations on NP management determined 5% lidocaine patches (local anesthetic drug) as first-line treatment and 8% capsaicin patches (local anesthetic drug) and BTX-A (peripherally acting muscle relaxant) as second-line treatments in patients with LNP [6].

The recommendation for the use of 5% lidocaine patches in LNP is weak with a moderate quality of evidence, according to the Grading of Recommendations Assessment, Development, and Evaluation (GRADE) system. However, topical lidocaine has an excellent safety profile and is commonly used in patients with several LNP syndromes [6]. Topically applied 8% capsaicin patches and BTX-A injections have a weak recommendation for use in patients with LNP with a high quality of evidence [6]. The clinical use of 8% capsaicin patches is limited by the special requirements of the application, i.e., application in specialized pain clinics by a well-trained team. Similar concerns are related to subcutaneous injections of BTX-A, whose availability is limited to specialized pain centers only [6].

Despite lacking robust evidence, several other topical treatments are currently used in clinical practice in patients with LNP, such as anesthetic drugs (capsaicin at low concentrations, ketamine), antiepileptics (gabapentin, phenytoin, cannabidiol), muscle relaxants (baclofen), antidepressants (amitriptyline, doxepin), mucolytics (ambroxol), antihypertensives (clonidine, prazosin), anti-inflammatory and antirheumatic drugs (diclofenac), antipruritics (menthol), antipropulsives (loperamide), and other analgesics (funapide) [17,18,19,20,21]. The scientific evidence for their use is inconclusive and comes mainly from case reports, case series, observational studies, or single RCTs. However, the low level of evidence does not exclude the possible effectiveness and beneficial analgesic effect in a given subject, particularly when other treatments fail, are contraindicated, or induce unacceptable side effects.

Topical analgesics utilized in clinical practice have undoubted advantages, including reduction in or absence of systemic side effects and reduced risk of overdose due to low or no systemic absorption from topical formulations, relatively few drug–drug interactions, ease of use and dose determination, avoidance of first-pass metabolism, improved patients’ compliance, and direct access to the target site and underlying pain mechanism. The disadvantages include the need for repeated applications per day for creams or gels, local side effects such as skin irritation, itch, or reddening, special requirements for 8% capsaicin patches and BTX-A application, and clinical usefulness limited to patients with LNP only [17,18,19,20,21]. Nevertheless, looking at molecular mechanisms, topical drugs act locally at peripheral mechanisms involved in NP generation, which is in line with current principles in pain medicine pointing out the need for personalized and mechanism-based approaches to pain management [22].

Molecules smaller than 500 dalton can easily penetrate the stratum corneum [23]. Nearly all active molecules are smaller than 500 dalton, and thus topically applied active molecules diffuse across the stratum corneum and influence structures in the epidermis (e.g., nociceptors, keratinocytes). Some active molecules penetrate deeper layers depending on the active molecule and the topical formulation. Active molecules of topical formulations may act on several distinct ion channels, receptors, proteins, or enzymes expressed by either neuronal or non-neuronal cells (Figure 1). The result of this process is the interruption of mutually intensifying stimulation loops, reduction in peripheral sensitization, and peripheral input, resulting in a reduction in NP, e.g., hyperalgesia and allodynia, in patients with LNP (Figure 2).

Since peripheral hyperexcitable cells of different types may become a target for topically administered analgesics, it is worth learning which receptors, ion channels, and/or enzymes involved in peripheral sensitization topical treatments may modulate directly or indirectly. In the subsequent sections, molecular/cellular mechanisms are presented, with a short description of the pathological changes occurring upon NP conditions, followed by preclinical and clinical data on topical treatments modulating a given ion channel, receptor, enzyme, or protein. Topical drugs utilized in subjects with LNP often exert multiple mechanisms of action, but to what extent each of these contributes to the analgesic effect observed in humans is not elucidated. This paper focuses mainly on topical treatments utilized in clinical practice. However, there are numerous preclinical trials pointing at the antinociceptive effect of substances administered topically in NP animal models, though they have not yet been introduced in clinical practice and are thus not included in this review [24,25]. 

## 3. Molecular/Cellular Mechanisms of Topical Treatments in Patients with Localized Neuropathic Pain

### 3.1. Treatments Acting on Voltage-Gated Sodium Channels

Voltage-gated sodium channels (Nav) are widely expressed in excitable cells, including peripheral and central neurons and cardiac and muscle cells, and are crucial for the initiation and propagation of action potentials. In sensory neurons, Nav determine the electrical excitability and play a key role in pain sensation by controlling afferent impulse discharges. Nav are expressed by non-excitable cells as well (e.g., keratinocytes, cancer cells), where they are involved in numerous biological processes [26]. The Nav family includes nine isoforms, 1.1 to 1.9, with different expressions within the peripheral and central nervous systems (CNS) and different physiological properties. In adulthood, Nav1.1, Nav1.6, Nav1.7, Nav1.8, and Nav1.9 are all expressed in primary sensory neurons and are involved in physiological nociception. However, they have different kinetics and distinct patterns of expression depending on the functional groupings of sensory neurons. Preclinical studies showed that some Nav subtypes with an abnormal function are linked to NP or other chronic pain states, i.e., Nav1.3, Nav1.6, Nav1.7, Nav1.8, and Nav1.9 [26,27]. Among these channels, subtypes Nav1.7, Nav1.8, and Nav 1.9 are highly expressed at nociceptors and thus may be good targets for topical analgesics. The most studied Nav1.7 and Nav1.8 differ with respect to their kinetic and voltage-dependent properties. Preclinical data showed that Nav1.7 serves as a threshold channel in peripheral sensory neurons; in turn, Nav1.8 contributes to repetitive firing and neuronal excitability. Therefore, the modulation of the latter could have a considerable impact on hyperexcitable neurons [28]. The role of Nav in nociception in humans has been confirmed in observations of individuals with loss-of-function mutations in the SCN9A gene coding for Nav1.7 and SCN11A coding for Nav1.9, exhibiting congenital insensitivity to pain with no other sensory abnormalities, except anosmia [29]. On the contrary, gain-of-function mutations in SCN9A in humans are linked to severe neuropathic pain states such as primary erythromelalgia and paroxysmal extreme pain disorder. Moreover, a common single-nucleotide polymorphism in SCN8A, SCN9A, SCN10A, and SCN11A genes coding for Nav1.6, Nav1.7, Nav1.8, and Nav1.9, respectively, may be linked to increased sensitivity to pain with neuropathic and non-neuropathic components in humans [29,30,31,32].

In models of neuropathic and inflammatory pain, extensive alterations in distribution, expression, trafficking, and/or biophysical properties of Nav subtypes have been observed. The expression and function of Nav may be regulated by intracellular signaling protein kinases (e.g., protein kinase A—PKA, protein kinase C—PKC, mitogen-activated protein kinase—MAPK), which in turn are subjects of injury-induced changes. The alterations in intracellular signaling pathways have been linked to sensory neuron hyperexcitability and pain in preclinical models [33].

In neuronal injury models, Nav1.7 and Nav1.8 were shown to accumulate at nerve injury sites [34]. However, expression of Nav in peripheral nerves may be up- or down-regulated depending on the NP animal models [27]. The up-regulation of Nav has been observed in humans with NP as well [35]. The accumulation and/or displacement of Nav at the nerve injury site and across the axon lead to membrane remodeling and modifications in Nav channels’ function, followed by changes in membrane electrical excitability (e.g., fast channel activation, increased Na+ current leading to hyperexcitation of peripheral nerve fibers) [28]. In models of inflammatory pain, the function of Nav is modulated by multiple inflammatory mediators such as prostaglandin E2 (PGE2), nerve growth factor (NGF), glial derived neurotrophic factor (GDNF), adenosine triphosphate (ATP), bradykinin, serotonin, adrenaline, and cytokines, e.g., tumor necrosis factor α (TNFα) and interleukin 1β (IL-1β), increasing the Na+ currents and excitability of peripheral nerve fibers in the injured area [27]. Taken together, the involvement of neuronal Nav in peripheral nerve hyperexcitability and pain behavior has been confirmed [36,37].

Furthermore, Nav1.1, Nav1.6, and Nav1.8 are expressed on both nerves and keratinocytes. These Nav on keratinocytes could possibly contribute to pain, due to their cross-talk with other epidermal structures. In patients with CRPS and PHN, the painful skin biopsies displayed Nav1.1, Nav1.2, and Nav1.8 immunolabeling, and substantially increased immunolabeling for Nav1.5, Nav1.6, and Nav1.7. Control skin exhibited immunolabeling for Nav1.5, Nav1.6, and Nav1.7 only [38]. Topically applied local anesthetics (lidocaine), antiepileptics (phenytoin), antidepressants (amitryptyline, doxepin), and other analgesics (funapide) and drugs (ambroxol) are thought to exert their analgesic effect in humans mainly via Nav inhibition [39,40,41,42,43,44,45,46].

#### 3.1.1. Lidocaine

Lidocaine, a local anesthetic agent having an aminoamide structure, is commonly used in regional anesthesia techniques. The main mechanism of action of lidocaine is through blockade of Nav channels. Lidocaine and other local anesthetics exert a frequency-dependent block (i.e., block intensity increases at higher action potential firing frequencies), bind preferably to the open or inactivated state of Nav, and decrease the intracellular influx of Na+, resulting in inhibition of the electrical impulse initiation and propagation [47,48]. Interestingly, Nav1.8 is about five times more sensitive to lidocaine than Nav1.7 or one of the other Nav subtypes. The modulation of Nav1.8 could have a significant impact on the hyperexcitable neurons, as Nav1.8 is responsible for repetitive firing and neuronal excitability [28,49,50]. In pain medicine, besides regional anesthesia techniques, topical and intravenous administration of lidocaine is commonly used, providing exposure of peripheral neurons to lidocaine at doses far below those that block nerve impulse propagation after perineural injection. However, even such low concentration of lidocaine is sufficient to induce an analgesic, but not an anesthetic, effect. The analgesic effect is related to the fact that the intensity of lidocaine blockade increases at higher action potential firing frequencies, observed in injured neurons expressing pathological Nav (i.e., frequency-dependent block) [28,47]. Topical application of lidocaine in NP animal models depresses ectopic activity in Aβ-, Aδ-, and C-fibers dose-dependently [51], and it reduces heat hyperalgesia in an animal model of HIV neuropathy [52]. In healthy subjects, differential effects of topical lidocaine on nociceptive Aδ- and C-fibers have been observed [39]. The increase in the sensory threshold and decrease in the evoked potentials amplitude were more prominent in C- than in Aδ-fibers [39]. Moreover, C-fibers are suggested to be a key player in chronic pain [53]. Taken together, preclinical and clinical data support topical administration of lidocaine in pain states, as it blocks the overactive Nav and suppresses the ongoing activity of rapidly firing neurons, which are thought to be the main mechanisms of NP [11,26,27,28].

In numerous preclinical in vitro and in vivo studies, lidocaine administered intravenously exhibited other mechanisms of action besides Nav blockade, which might contribute to antinociception and local anesthetic activity. Whether these mechanisms play a role in pain relief following topical application of lidocaine is still not clear:Blockade of muscarinic acetylcholine receptors (mAChR) at concentrations 1000-fold lower than needed for Nav blockade [54];Blockade of N-methyl-d-aspartate receptors (NMDAR) and inhibition of glutamate release from nerve terminals at clinically relevant plasma concentrations after intravenous administration [55,56,57];Inhibition of hyperpolarization-activated cyclic nucleotide-gated (HCN) channels at concentrations which block Nav1.8 [58];Inhibition of Toll-like receptor 4 (TLR4) at concentrations which block Nav1.8 [59];Blockade of voltage-gated calcium channels (VGCCs), but at doses 100-fold higher than needed for Nav blockade [60,61];Blockade of several types of potassium channels: voltage-gated (Kv), tandem pore domain (K2P), and inwardly rectifier (Kir), at concentrations several-fold higher than needed for Nav blockade [62,63,64];Desensitization of transient receptor potential ankyrin 1 (TRPA1) ion channels [65]; however, activation of transient receptor potential vanilloid 1 (TRPV1) and TRPA1 has been observed in rodents as well, which may contribute to lidocaine-induced neurotoxicity [66,67];Inhibition of acid-sensing ion channels (ASIC), but at doses 100-fold higher than needed for Nav blockade [68];Inhibition of P2X purinoceptors receptor 7 (P2X7) subunits, expressed in microglia, but the exact mechanism of interaction between lidocaine and the purine receptor remains unclear [69];Suppression of NGF/tropomyosin receptor kinase A (TrkA) signaling due to the structural similarity of Nav and TrkA [70];Anti-inflammatory properties—reduction in neuroinflammation, probably via G protein-coupled receptors (GPCR), inhibition of granulocytes migration and microglial activation, reduced release of inflammatory cytokines TNFα, IL-6, and IL-1β from microglia and macrophages, reduced sensitization of peripheral nerve endings; moreover, prolonged (hours) exposure of cells to lidocaine enhances its effects on GPCR signaling [71,72,73,74,75];Modulation of glycinergic pathways—lidocaine metabolite N-ethyl glycine inhibits spinal glycine transpor ter (GlyT1) which increases levels of glycine in the serum and spinal cord, resulting in antinociception in an NP model [76,77].

In clinical practice, several formulations of topical lidocaine producing local analgesia have been used to date such as EMLA patches or cream (eutectic mixture of local anesthetics—lidocaine 2,5%, prilocaine 2,5%), 2–11% lidocaine gel, cream, or spray, 7% lidocaine with 7% tetracaine cream, and 5% lidocaine patches. However, only 5% lidocaine patches are registered and recommended in patients with LNP [5,6,78]. Topical 5% lidocaine patches were first registered in the USA in 1999 and, since then, have been commonly used in patients with LNP, such as PHN, PDN, post-traumatic, and post-surgical nerve injury [78]. In clinical trials conducted in patients with PHN, 5% lidocaine patches were applied topically over the painful area(s), which reduced the intensity of all neuropathic pain characteristics measured on the Neuropathic Pain Scale (NPS), and allodynia [79,80]. A clinical study showed that treatment with topical lidocaine decreased activation of specific regions in the CNS, measured with functional magnetic resonance imaging, implying a close correlation between peripheral input and central pain processing [81]. To date, the clinical use of 5% lidocaine patches in LNP is supported by several RCTs and reviews [82,83,84].

#### 3.1.2. Phenytoin

Phenytoin is a hydantoin derivative, a first-generation anticonvulsant drug, effective in the treatment of generalized tonic–clonic seizures, complex partial seizures, and status epilepticus. Phenytoin non-selectively blocks voltage-dependent Nav, leading to a reduction in the firing of neurons, and resulting in anticonvulsant and anti-neuropathic properties [40]. It is suggested that phenytoin blocks Nav poorly at slow firing rates but suppresses the high-frequency repetitive firing [40]. Phenytoin (IC_50_ = 40 μM) has six times stronger Nav binding activity than lidocaine (IC_50_ = 240 μM) [85]. The non-selective Nav blockade by phenytoin is thought to be of key importance in LNP management. However, other preclinical studies reveal other mechanisms of action, which might potentially contribute to its antinociceptive properties: Blockade of L-type VGCCs, observed in smooth muscle preparations as the inhibition of their spontaneous activity [86];Potentiation of gamma-aminobutyric acid (GABA)-induced currents through modulation of the gamma-aminobutyric acid A receptor (GABAAR) in cultured rat cortical neurons [87];Anti-inflammatory properties—reduction in tissue edema, decrease in inflammatory cell infiltration, and increase in epidermal growth factor, vascular endothelial growth factor, and transforming growth factor-β (TGFβ) in a rat model of wound healing [88];Antinociceptive effect in inflammatory pain models [89].

In pain medicine, systemic phenytoin is recommended and utilized in patients with trigeminal neuralgia as monotherapy or add-on therapy [90] and in patients with NP exacerbations [91,92]. To date, various topical formulations containing phenytoin have been used to treat diabetic wounds and ulcers, and other difficult-to-treat wounds to enhance their healing [93]. Data from preclinical studies support phenytoin effectiveness in wound healing [88]. However, the evidence on its efficacy in clinical practice is still inconsistent [94]. The long history of topical phenytoin utilization in wound healing without any signs of dermatotoxicity and its possible mechanisms of action lead to the idea of topical phenytoin application in patients with LNP. To date, observational studies on the efficacy and safety of topical phenytoin up to 30% have been performed in more than 100 patients with LNP of several etiologies including hernia pain [95,96,97,98,99,100,101]. In an observational study describing 70 neuropathic pain patients treated with phenytoin 5% and 10% creams, 70% of these patients experienced at least 50% pain relief [95]. In 16 patients, phenytoin plasma levels were measured. No phenytoin plasma levels were detected, even after the application of 6.7 g of the phenytoin 10% cream in one case. Since phenytoin cream can provide pain relief around 15 min after application and polyneuropathic pain is usually symmetrical in location (both feet and/or lower legs) and intensity, a single-blind placebo-controlled test (SIBRET) was developed [96]. On one area, placebo cream consisting of the base (placebo) cream was applied, and on the other area, phenytoin 10% cream was applied. Responders experienced more pain relief in the phenytoin 10%-applied area than in the placebo cream-applied area within 30 min after application. In the first study evaluating SIBRET, of the 21 patients, 15 were classified as responders [96]. The mean pain reduction after 30 min as measured with the 11-point numerical rating scale in the phenytoin 10% cream area was 3.3 (SD: 1.3) and in the placebo cream area 1.2 (SD: 1.1). The pain-relieving effects of phenytoin cream compared with placebo cream were confirmed by a study examining the double-blind placebo-controlled response test (DOBRET) [98]. Six out of 12 NP patients were classified as responders. All responders had at least 30% pain reduction, and four out of six had at least 50% pain reduction in the phenytoin 10% cream-applied area.

Future RCTs, such as a triple cross-over trial evaluating 10% and 20% phenytoin creams compared with placebo cream, in painful chronic idiopathic axonal polyneuropathy (CIAP) patients will give more insight into the pain-reducing effect of topical phenytoin [102]. Additionally, the predictive value of DOBRET for the maintenance of pain relief with topical phenytoin will be explored.

#### 3.1.3. Ambroxol

Ambroxol is a mucolytic drug with several properties including secretolytic and secretomotor actions. Data from preclinical studies have shown its potent local anesthetic activity due to Nav1.7 and Nav1.8 inhibition [41] and anti-inflammatory activity due to reduction in proinflammatory cytokines such as IL-1β, IL-6, IL-17, IL-22, IL-23, TGFβ, and TNFα [103]. Ambroxol is a very potent Nav blocker, approximately 40 times stronger than lidocaine [104], and probably preferentially blocks the subtype Nav1.8, which is responsible for repetitive firing and neuronal excitability [28,41]. Ambroxol has been studied in patients with different NP syndromes, including trigeminal neuralgia, and has shown relevant pain relief following topical administration of 20% ambroxol cream [42,105]. In a case series of CRPS patients, topical 20% ambroxol reduced spontaneous pain, edema, allodynia, hyperalgesia, and skin reddening and improved motor dysfunction and skin temperature [106]. The evidence for topical ambroxol in NP comes from observational studies only but might be beneficial in some patients with LNP.

#### 3.1.4. Amitriptyline

Amitriptyline is a tricyclic antidepressant (TCA), recommended and used in oral formulations as a first-line treatment in patients with peripheral and central NP [5,6]. The analgesic mechanism of action of TCAs relies on targeting the multiple sites in the nociceptive system, observed in experimental studies, either in the CNS or in the periphery:Inhibition of neuronal reuptake of noradrenaline and serotonin in the spinal cord [44];Increase in dopamine concentration in the spinal cord [44];Activation of the locus coeruleus in the posterior brainstem and activation of the descending noradrenergic endogenous antinociceptive system [44];Blockade of Nav—most potent for Nav1.7, Nav1.8, and Nav1.9 in use-dependent manner in the range of therapeutic plasma concentrations for the treatment of depression and NP [43,107,108];Blockade of NMDAR in cultured rat brain neurons [109];Activation of Kv channels in vivo [110];Increase in GABAAR and gamma-aminobutyric acid B receptor (GABABR) activation [111,112];Indirect involvement of opioid system, probably through endogenous opioid release [113];Activation of TRPA1 channels and subsequent probable desensitization contributing to the analgesic effect [108];Down-regulation of α1 adrenergic receptor (α1-AR) in the rat brain [114];Blockade of serotonin, histamine, and muscarinic receptors in the peripheral nervous system [115,116];Inhibition of the neuronal uptake of adenosine [117];Inhibition of the production of nitric oxide and PGE2 in synovial tissue cultures [118].

The results from preclinical studies suggest that the main antinociceptive mechanism of action of amitriptyline is related to the potent blockade of Nav1.7, Nav1.8, and Nav1.9, since peripheral injection of amitriptyline induced local anesthesia lasting longer than that after bupivacaine injection in rats [119]. The antiallodynic effect of peripherally administered amitriptyline as a subcutaneous injection has been shown in a rodent model of streptozotocin (STZ)-induced diabetic neuropathy [120]. Another preclinical study showed that topical application of amitriptyline was more potent than lidocaine at the same concentrations in providing cutaneous analgesia in rats [121]. Taken together, the preclinical evidence shows that topical application of amitriptyline exerts antinociceptive and antiallodynic effects in NP and acute pain models [108,119,120,121]. In healthy volunteers, local amitriptyline injection was evaluated as an ulnar nerve blockade, though it was found to be less effective than bupivacaine injection [122]. Topical application of amitriptyline in 50 mmol/L and 100 mmol/L solutions was significantly more effective in providing cutaneous analgesia than placebo in healthy human volunteers, with some subjects having a complete analgesia lasting several hours [123]. Topical applications of amitriptyline in several concentrations have been tested in patients with different LNP syndromes; however, results were ambiguous. The tendency was that the higher the concentration, the more pronounced the effect. In most clinical trials, the combination of amitriptyline and ketamine in a 2:1 ratio was evaluated [124]. Some clinical trials of topical administration of 1–5% amitriptyline in combination with ketamine and in various NP syndromes (PHN, PDN, post-traumatic NP, painful peripheral neuropathy) did not show statistical significance in pain relief [125,126,127]. On the other hand, a clinical trial following an enrichment design evaluating topical amitriptyline 4% and ketamine 2% in PHN patients showed a statistical significance compared with topical placebo [128]. Data from case reports indicate pain-relieving effects of topical 5–10% amitriptyline applications in patients with PDN, CIAP, CRPS, and post-traumatic NP [124,129,130]. However, a systematic review assessing the analgesic effect of topical amitriptyline in patients with LNP, published in 2015, concluded that data from RCTs evaluating the lower concentrations of amitriptyline (up to 5%) do not support topical amitriptyline for LNP [131]. Nevertheless, two recently published clinical trials (one case series, one pilot study) showed some analgesic effectiveness of 10% topical amitriptyline in patients with CIPN, which suggests its possible usefulness in clinical practice, particularly in patients with refractory LNP [108,132].

#### 3.1.5. Doxepin

Doxepin is a TCA which may be administered topically in patients with NP. Doxepin is supposed to share the same analgesic mechanisms of action with amitriptyline. The analgesic efficacy of topical 3.3% doxepin alone or in combination with 0.025% capsaicin was shown in an RCT in 200 patients with chronic NP and CRPS [133]. Recently, the analgesic effect of topical 5% doxepin was described in a case report of a pediatric patient with lymphoblastic leukemia and severe NP following antifungal treatment [134]. More evidence on the analgesic efficacy of locally administered doxepin comes from two RCTs in cancer patients with oral pain due to mucositis, following radiotherapy. The mouthwash containing 25 mg doxepin in 5 ml water significantly reduced oral mucositis pain during the first 4 h after administration; however, the effect was minimal [135,136].

Data from preclinical studies suggest that various TCAs may differentially block Nav. In animal models, amitriptyline, doxepin, and imipramine applied perineurally were superior to bupivacaine in blocking nerve impulse propagation in a rat sciatic nerve. Trimipramine and desipramine were less effective than bupivacaine, and nortriptyline, protriptyline, and maprotiline were inferior; therefore, these probably will not induce an antinociceptive effect when applied topically. However, it is not clear whether these differences are associated with different activities at Nav or with different penetrations of TCAs into the neuronal membrane in animal models [137].

#### 3.1.6. Funapide

Funapide, also coded as TV-45070, is a selective blocker of Nav1.7, applied topically in a cross-over RCT in patients with PHN. No statistical difference was observed between treatments for the primary endpoint, i.e., the difference in change in mean daily pain score from baseline compared with the last week. However, the proportion of patients with 50% pain reduction at week 3 was greater on topical funapide than on topical placebo (26.8% vs. 10.7%, *p* = 0.0039). Moreover, 63% of patients with the R1150W polymorphism of the Nav1.7 coding gene vs. 35% of wild-type carriers had a 30% reduction in mean pain score on TV-45070 at week 3 [46].

#### 3.1.7. Other Drugs—Nonsteroidal Anti-Inflammatory Drugs, Opioids, Ketamine, Menthol, Cannabidiol

Other drugs which might block Nav, observed at in vitro peripheral nerves, include nonsteroidal anti-inflammatory drugs (NSAIDs), opioids, α2 adrenergic receptors (α2-AR) agonists, and plant-derived compounds, recently reviewed by Kumamoto [138]. Whether and to what extent the Nav blockade of these drugs is responsible for their analgesic effect observed in humans remain unclear. In vitro opioid studies examining morphine, tramadol, fentanyl, sufentanil, and buprenorphine showed a decrease in action potential amplitudes and conduction via direct action on Nav and Kv channels in frog sciatic nerves [138], and on unmyelinated mouse C-fibers [139]. Moreover, loperamide inhibited Nav1.7 and Nav1.8 as well in an in vitro study [140].

NSAIDs such as diclofenac, aceclofenac, indomethacin, tolfenamic, and flufenamic acid reduced action potential amplitudes and inhibited nerve conduction through Nav inhibition in frog sciatic nerves in a concentration-dependent manner, probably due to their chemical structure being similar to local anesthetics [138]. A similar inhibitory effect on the action potential was observed after application of clonidine (α2-AR agonist) on frog sciatic nerves, which was attributed to Nav blockade [138]. Moreover, ketamine was shown to directly inhibit Nav in neuroblastoma cells [141], which may account for the analgesic effect of ketamine as adjuvant to local anesthetic agents. Additionally, menthol selectively blocks Nav1.8 and Nav1.9, which has been shown in rat DRG neurons [142]. In preclinical studies, it has been shown that cannabidiol (CBD) may directly interact with Nav and Kv channels [143]. In an RCT, testing topical CBD oil in 29 patients with peripheral neuropathic pain compared to placebo, CBD oil showed a more pronounced pain-reducing effect than placebo, but the authors did not discuss the possible Nav blockade as the mechanism of action of CBD applied topically in the painful area [144].

### 3.2. Treatments Acting on Transient Receptor Potential Channels

The transient receptor potential channels (TRP) family is the largest group of ion channels. TRP can be classified into six subfamilies, based on their structure: TRPA (ankyrin), TRPC (canonical), TRPM (melastatin), TRPML (mucolipin), TRPP (polycystin), and TRPV (vanilloid). TRP are widely distributed across tissues, such that every cell in the body likely expresses one or more subtypes [145]. Some members of the TRP family deserve special attention in pain research such as TRPV1-4, TRPM8, and TRPA1. They are expressed at nociceptors and convey thermal, chemical, and mechanical stimuli, and they are involved in the development and maintenance of chronic pain [146]. Moreover, the skin expresses several TRP subtypes (i.e., TRPV1-4, TRPA1, TRPM8), involved in skin biology under physiological (e.g., sensory function, nociception, epidermal homeostasis, inflammation) and pathological conditions (e.g., inflammatory and neuropathic pain, dermatitis, itch), as reviewed by Caterina and Pang [145].

#### 3.2.1. Treatments Acting on Transient Receptor Potential Vanilloid 1

TRPV1 channels are polymodal receptors expressed mainly at C-fiber nociceptors in the periphery, involved in membrane depolarization and controlling cytoplasmic Ca2+. TRPV1 can also be found in the central nervous system and in several non-neuronal cells as well (i.e., keratinocytes, bladder transitional epithelial cells, smooth muscle cells; kidney, lung, testis, uterus, spleen, liver, and pancreas cells; granulocytes, lymphocytes, and macrophages) [145,146,147]. TRPV1 may be activated by a wide spectrum of physical and chemical stimuli such as heat (>43 °C), protons (pH < 6.5), endogenous substances such as endocannabinoids (i.e., anandamide, N-arachidonoyl dopamine) [146,148], exogenous compounds such as capsaicin, phyto-cannabinoids, and some animal toxins [149,150,151]. TRPV1 is colocalized with TRPA1 in DRG and trigeminal neurons, and there are direct and/or indirect functional interactions between TRPA1 and TRPV1 involved in nociception [152]. Available data suggest that TRPV1 plays an important role in the pathomechanism of neuropathic and inflammatory pain [153,154,155,156,157,158]. Nerve injury decreases TRPV1 levels in injured neurons and increases expression of TRPV1 in uninjured or spared neurons, observed after nerve ligation/transection in animal models. [159,160]. In humans, accumulation of TRPV1 and TRPV3 in spared peripheral axons in patients with brachial plexus avulsion and a reduction in TRPV1 levels in nerve fibers in diabetic neuropathy skin were observed [161].

After injury, numerous signaling molecules such as bradykinin, histamine, proinflammatory cytokines (TNFα), glutamate, and NGF are released from damaged nerves, Schwann cells, and immunocompetent cells. These molecules activate and/or sensitize neuronal TRPV1, contributing to thermal and mechanical hypersensitivity in NP and inflammatory pain [162,163,164]. The role of TRPV1 in pathological nociception has been confirmed in patients with small fiber neuropathy, in whom a statistically significant increase in TRPV1 expression on epidermal keratinocytes was reported [158]. In turn, in patients with PDN, a decrease in TRPV3 in the skin was observed [161].

##### Capsaicin

Capsaicin is a highly selective agonist of the TRPV1 channels utilized in clinical settings either in low (<0.1% cream) or in high (8% patches) concentrations in patients with LNP [165]. Although capsaicin is the potent activator of TRPV1, its long-term analgesic effect relies on the massive intracellular influx of ions (Ca^2+^, Cl^−^) after TRPV1 activation and subsequent intracellular changes. To specify, capsaicin causes a three-fold increase in the permeability to Ca^2+^ of the ion channels coupled with TRPV1. The increased influx of Ca^2+^ into the cell and the release of Ca^2+^ from the endoplasmic reticulum activate proteases and initiate subsequent damage of the cytoskeleton and mitochondria. This leads to the de-functionalization of hyperexcitable TRPV1, or a temporary destruction of peripheral nerve endings [147]. Clinical evidence supports only 8% capsaicin patches in patients with LNP [5,6,166,167,168], whereas the evidence for the low concentration capsaicin is inconclusive [5,6]. Single application of an 8% capsaicin patch in subjects with LNP brings significant pain relief within 1–2 weeks, resulting from de-functionalization and temporary destruction of nerve endings in the area of the patch application. Nerve endings regenerate after an average of 3 months, which may be associated with pain recurrence, and in this case, the application of the patch can be repeated [167,168]. The application of a high-concentration capsaicin patch may induce or aggravate severe burning pain. The recommendation before using the 8% capsaicin patch is to apply a local anesthetic such as EMLA cream at the corresponding area of LNP to diminish the risk of severe burning pain. During the treatment procedure, blood pressure monitoring should be performed, and nitrile gloves, a face mask, and protective glasses must be worn in a well-ventilated treatment area. Thus, the 8% capsaicin patch application can only be performed in specialized pain clinics. Nevertheless, clinical data support the effectiveness of 8% capsaicin patches in patients with several LNP syndromes, such as PHN, HIV-associated neuropathy, CIPN, and PDN [166].

##### Other Drugs—NSAIDs, Cannabinoids

Other drugs acting via TRPV1 include NSAIDs such as diclofenac, ketorolac, and xefocam. These molecules applied topically in rats inhibited pain behavior, most probably by inhibition of TRPV1 and TRPA1 channels [169]. Endocannabinoids and phyto-cannabinoids may exert their antinociceptive effect via TRPV1 activation, as was shown in preclinical studies. Their mechanisms of analgesic action may be similar to capsaicin, i.e., initial activation of TRPV1, followed by its desensitization [148,150,151].

#### 3.2.2. Treatments Acting on Transient Receptor Potential Melastatin 8

TRPM8 is expressed in the skin at nociceptors and keratinocytes and is responsible for detection of mild cold stimuli. TRPM8 is activated by temperatures below 28 °C, and by menthol and other cooling agents. The relationship of TRPM8 to pain is more complex. TRPM8 co-expression with TRPV1 in nociceptive neurons may contribute to hypersensitivity to cold stimuli in inflammatory and neuropathic pain models. In contrast, TRPM8 stimulation may attenuate pain sensitivity as well [170].

##### Menthol

Menthol is the cooling natural molecule of peppermint, commonly used in medicinal preparations for the relief of acute and inflammatory pain in sports injuries, arthritis, and other painful conditions in humans. Menthol’s main mechanism of analgesic action is attributed to TRPM8 activation [171]. However, other analgesic mechanisms of action are suggested, as shown in preclinical studies:Inhibition of human TRPA1 channels in vitro [172];Inhibition of VGCCs in human neuroblastoma cells [173];Activation of human recombinant GABAAR expressed in Xenopus oocytes [174];Selective blockade of Nav1.8 and Nav1.9 in rat DRG neurons [142];Inhibition of human recombinant nAChR [175].

In animal pain models, l-menthol (predominant isomer in menthol formulations) effectively reduced pain behavior induced by chemical stimuli such as capsaicin, noxious heat, and inflammation. Moreover, in animal models, the role of TRPM8 as the main mediator of menthol-induced analgesia of acute, neuropathic, and inflammatory pain has been confirmed [171]. Clinical studies are sparse. In eight LNP patients, topical application of menthol attenuated cold allodynia [176]. One case report also showed a beneficial analgesic effect of topically applied peppermint oil in a patient with PHN [177]. Historically, the first description of peppermint oil for NP comes from a letter to the editor of The Lancet in 1870 in which Dr. A. Wright reported on peppermint oil being used to treat “facial neuralgia” in China and in his own clinical practice [178].

On the other hand, a relatively high concentration of menthol (>30%) applied topically induces cold pain and hyperalgesia in healthy volunteers [179]. Therefore, in clinical practice, lower concentrations of menthol are used.

### 3.3. Treatments Acting on Voltage-Gated Calcium Channels

VGCCs can be classified as L, N, P/Q, R, and T and are distinguished by their different sensitivities for pharmacological agents and their channel conductance kinetics based on their voltage activation properties. Different VGCC isoforms show distinct cellular and subcellular distributions to play specific functional roles. VGCCs are widely distributed in neuronal and non-neuronal cells. Studies confirmed expression of L-type VGCCs in excitable cells [180] and in epidermal keratinocytes, where they play a role in skin barrier homeostasis [181]. In nociceptive neurons, neurotransmitters such as glutamate, substance P (SP), and calcitonin gene related peptide (CGRP) are released after activation of VGCCs, mainly the L, N, and P/Q types [182]. In turn, T-type VGCC is associated with the regulation of neuronal excitability and the activity of the T type is increased in the central terminal of nociceptors in NP states, such as traumatic nerve injury, PDN, or CIPN [183,184]. A recent preclinical in vivo study showed the expression of functional N-type VGCCs (Cav2.2) in skin nociceptors, being responsible for release of inflammatory signals and being involved in neurogenic thermal hyperalgesia [185]. Due to the crucial role of VGCCs in pain processing and their distribution in keratinocytes and neuronal cells, VGCCs may be targeted not only by systemic treatments but also by topical treatments.

#### 3.3.1. Gabapentin

Gabapentin is an antiepileptic and anxiolytic agent recommended for oral intake as the first-line treatment in patients with NP [5,6]. Gabapentin is an analogue of GABA; however, it does not influence GABAR or GABA synthesis and uptake. The main mechanism contributing to gabapentin’s analgesic effect in NP states is related to interactions with α2δ-1 subunits of VGCCs, and the subsequent reduction in Ca^2+^ influx and transmitter release [186]. In preclinical studies, other mechanisms of action contributing to the antinociceptive properties of gabapentin are suggested:Blockade of human recombinant NMDA in a concentration-dependent manner in vitro [187];Attenuated cytokines production, COX-2 expression, and PGE2 levels in animal model of ocular inflammation by topical gabapentin [188];Activation of human Kv channels in vitro [189].

Preclinical studies showed the antinociceptive effect of topically administered gabapentin in animal models of peripheral nerve injury (10% gel) [190], CIPN (10% gel) [191], and formalin-induced pain (1–10% cream) [192]. In clinical settings, cream containing gabapentin showed a beneficial analgesic effect in patients with several NP states [193] and in patients with local or generalized vulvodynia [194]. In patients with LNP, 6% topical gabapentin has been utilized in combination with other compounds providing pain relief [195,196]. The evidence for the effectiveness of topical gabapentin in patients with NP is limited to single observational studies and case reports; moreover, the exact mechanism of analgesic action is not clear. Theoretically, gabapentin may influence VGCCs, NMDA, Kv, and inflammatory mediators, leading to reduced neuronal hyperexcitability and antinociception, but it needs thorough evaluation in preclinical and clinical studies.

#### 3.3.2. Other Drugs

Whether lidocaine, phenytoin, and menthol exert their analgesic mechanism of action via VGCC blockade upon topical application in animal models and in humans is unclear; however, in preclinical studies, these drugs influenced the VGCCs, as discussed earlier [60,61,86,173].

### 3.4. Treatments Acting on N-methyl-D-Aspartic Acid Receptors

NMDAR is a receptor for the excitatory neurotransmitter glutamate, which is released upon activation of nociceptive afferents, especially of the unmyelinated C-fibers. It is well known that NMDAR activation plays a key role in the central sensitization of spinal nociceptive neurons, resulting in allodynia, hyperalgesia, and NP [197]. There is considerable evidence as well for the modulatory role of glutamate of the NMDAR and non-NMDA glutamate receptors in peripheral nociception, as these receptors are present in the peripheral terminals of C-fibers [198]. Upon stimulation, peripheral C-fibers release glutamate, SP, and CGRP [199], which cause neurogenic inflammation and, in a paracrine manner, modulate neuronal excitability via receptors expressed at nearby nociceptors [200] and contribute to tactile hypersensitivity in animal models [201]. The role of peripheral NMDAR has been confirmed in human studies as well in which the local inhibition of peripheral NMDAR prevented the development of secondary hyperalgesia by a peripheral mechanism of action [202].

#### 3.4.1. Ketamine

Ketamine is an anesthetic and analgesic agent, usually administered intravenously. Topical application of ketamine for LNP relies mainly on blockade of the peripheral NMDAR, α-amino-3-hydroxy-5-methyl-4-isoxazolepropionic acid receptor (AMPAR), and metabotropic glutamate receptor (mGluR) in a non-competitive fashion and inhibition of the release of glutamate [203].

Furthermore, in preclinical studies, ketamine shows influence in a direct or indirect manner on several ion channels and receptors, which may be involved in peripheral nociception by influence of:Blockade of Nav in neuroblastoma cells [141];Re-sensitization of opioid receptors (OR)—μ opioid receptors (MOR) and δ opioid receptors (DOR) [204,205];Activation of GABAA receptor in an anesthetic model in mice [206];Inhibition of L-type VGCCs in smooth muscle cells [207];Blockade of HCN1 channels [208];Inhibition of nAChR in human neurons [209] and mAChR in mice [210];Reduced expression of TLR4 and proinflammatory cytokines release from immune cells [211].

In clinical trials, topical ketamine is commonly used in combination with other drugs, showing a beneficial analgesic effect in patients with PHN, CIPN, PDN, and CRPS:Ketamine (2%), amitriptyline (4%) [128,212,213,214];Ketamine (1.5%), baclofen (0.8%), amitriptyline (3%) [215];Ketamine (10%), baclofen (2%), gabapentin (6%), amitriptyline (4%), bupivacaine (2%), nifedipine (2%), clonidine (0.2%) [195];Ketamine, pentoxifylline, clonidine, dimethyl sulfoxide [216];Ketamine (5%), clonidine (0.5%), gabapentin (6%) [196].

Only a few studies have described clinical results after topical application of ketamine alone. The topical application of ketamine in different concentrations (0.5–5%) showed no beneficial analgesic effect in RCTs in patients with PHN, PDN, and post-traumatic NP [125,126,217,218]. However, other clinical studies showed good results in patients with PHN treated with low-concentration topical 0.5% ketamine [219] and in a case series including patients with CRPS, lumbar radiculopathy, and PHN [220]. Topical ketamine 10% reduced allodynia and hyperalgesia in patients with CRPS in one RCT [221] and had a beneficial effect reported in several case series, case reports, and retrospective studies [222,223,224,225].

#### 3.4.2. Other Drugs

Other drugs acting via NMDAR blockade in preclinical studies include:Antidepressants (e.g., amitriptyline), though this effect has been observed in cultured rat brain neurons only [109];Diclofenac, providing an antinociceptive effect after topical administration in rats [226];Lidocaine at clinically relevant plasma concentrations [55,56,57].

It is still unclear to what extent topically applied amitriptyline, diclofenac, and lidocaine induce analgesia via NMDAR blockade in humans.

### 3.5. Treatments Acting on Cyclooxygenase-2

Peripheral nerve injury induces Schwann cells and macrophages to produce and release cytokines and arachidonic acid and its derivates, mainly prostaglandins (PGs), via COX-2 induction. After an injury, PGs may be synthetized not only in the invaded immune cells but also in neuronal and glial cells. PGs play an important role in regulating the function of peripheral sensory nerves in paracrine and autocrine manners elucidated in NP models [227]. Preclinical studies reveal that PGE2, via its EP receptor expressed on the neuronal membrane, can modulate the excitability of peripheral nerve endings. PGE2 sensitizes several ion channels and receptors (TRPV1, Nav1.7, Nav1.8, Nav1.9, VGCCs, P2X3) and down-regulates Kv, which results in enhanced Na^+^ currents and Ca^2+^ influx and reduced K^+^ currents, resulting in peripheral hyperexcitability [155].

#### Nonsteroidal Anti-Inflammatory Drugs—Diclofenac

Topically administered NSAIDs may interfere with the nociceptive pathway by their ability to decrease the synthesis of proinflammatory PGs through inhibition of cyclooxygenase COX-2. Preclinical studies showed that locally administered diclofenac may also act on several ion channels (Nav, TRPV1, TRPA1, TRPM3, K^+^, NMDAR) at peripheral neurons, resulting in antinociception [138,169,226,228,229]. Peripheral antinociception induced by diclofenac may rely on release of noradrenaline and interaction with α1, α2C, and β-adrenoreceptors [230]. Additionally, the opioid system may be involved as well by indirect activation of the κ opioid receptors (KOR), probably by release of endogenous opioids such as dynorphins [231]. Diclofenac also blocks L-type VGCCs, but this effect was observed in neonatal rat ventricular myocytes and whether this mechanism is involved in antinociception is unknown [232]. In a cross-over placebo-controlled RCT with 28 PHN and CRPS patients, gel containing 1.5% diclofenac gave more pain relief than placebo cream [233]. Case reports describe a positive analgesic effect of topically administered ibuprofen and ketoprofen in combination with other agents in patients with LNP [234]. However, topical NSAIDs are not widely used in patients with LNP and are rather recommended and commonly used in patients suffering from pain with a predominant inflammatory component [83].

### 3.6. Treatments Acting on Gamma-Aminobutyric Acid Receptors

GABA is the major inhibitory neurotransmitter in the adult mammalian nervous system and exerts inhibitory action via specific receptors named GABAAR and GABABR. GABAAR are GABA-gated chloride channels located in post-synaptic membranes, whereas GABABR are G protein-coupled receptors located both in pre- and post-synaptic membranes [235]. GABA receptors (GABAR) are expressed on neuronal cells, either centrally or peripherally, and on CNS glial cells and myelin-producing Schwann cells [236]. In peripheral neurons, GABAR activation by agonists results in inhibition of signal transmission, due to an intracellular increase in K^+^ and a decrease in Ca^2+^ ions [237,238]. In the periphery, the GABABR are found in cutaneous layers on nerve endings [238]. In preclinical studies, it has been revealed that immune cells (macrophages, neutrophils, and lymphocytes) may express components of the GABAergic system as well [239]. GABA signaling is involved in the modulation of the immune response through reduction in proinflammatory cytokine production by down-regulation of signaling pathways (e.g., MAPK) [239]. Skin cells such as keratinocytes and fibroblasts express GABABR as well, which are involved in skin barrier homeostasis [240] and inflammatory diseases [241].

#### 3.6.1. Baclofen

Baclofen is a selective agonist of GABABR, traditionally used as a systemic treatment for spasticity. Topical application of baclofen in patients with LNP may potentially reduce pain due to GABABR activation, subsequent inhibition of neural transmission, and probably attenuation of local inflammation [239]. In preclinical studies, single subcutaneous injection of baclofen 0.01% reduced thermal hyperalgesia in mice with mixed nociceptive and neuropathic pain [242]. In humans, topical baclofen 5% successfully relieved NP due to acromegaly [243] and spinal cord injury [244,245]. In clinical practice, baclofen was more commonly used in combination with other topical agents:Baclofen (0.8%), amitriptyline (3%), ketamine (1.5%) [209];Baclofen (2%), ketamine (10%), gabapentin (6%), amitriptyline (4%), bupivacaine (2%), nifedipine (2%), clonidine (0.2%) [195];Baclofen (5%), palmitoylethanolamide (1%) [246];Baclofen, diclofenac, ibuprofen, cyclobenzaprine, bupivacaine, gabapentin, pentoxifylline [234].

In the aforementioned clinical studies, topical baclofen as monotherapy or add-on therapy was shown to be effective in pain relief; however, evidence is limited to case reports or single RCTs.

#### 3.6.2. Other Drugs

Other drugs acting on GABAR in preclinical studies include:Antidepressants (amitriptyline, fluoxetine)—their antinociceptive effect has been observed after intraperitoneal administration in rats [111];Ketamine, acting as an agonist at GABAAR, revealed in an anesthetic mouse model [206];Phenytoin, potentiating GABA-induced currents in cultured rat cortical neurons through modulation of GABAAR [87];Menthol, increasing GABA-induced currents by activation of human recombinant GABAAR expressed in Xenopus oocytes [174].

It is still unrevealed whether antidepressants, ketamine, phenytoin, and menthol induce analgesia via GABAR upon topical application in subjects with LNP.

### 3.7. Treatments Acting on α Adrenergic Receptors

Preclinical models of peripheral nerve injury showed that proinflammatory cytokines and growth factors can increase the expression of α1-AR on nociceptive afferent fibers and DRG that survive the nerve damage, and on immune cells and keratinocytes [247,248]. In turn, activation of α1-AR on immune cells and keratinocytes by noradrenaline may trigger further release of growth factors and inflammatory mediators, perpetuating the cycle and contributing to inflammation and pain [247,248]. Evidence of the aberrant adrenergic influence on nociception in NP comes from clinical observations as well. In some CRPS patients, α1-AR were up-regulated in the epidermis and on dermal nerve fibers [249,250]. In these patients, intradermal injection of α1-AR agonist phenylephrine evoked prolonged pain and more pronounced pinprick hyperalgesia in comparison to CRPS patients with less expressed α1-AR [249,250]. On the other hand, α2-AR are inhibitory G protein-coupled receptors [251,252]. α2-AR are expressed in the brain, spinal cord, and DRG and on nociceptors in the epidermis. Activation of these receptors likely decreases levels of adenylate cyclase and cyclic adenosine monophosphate (cAMP), resulting in decreased neurotransmitter release and reduced excitability of nociceptors, expressed in reduced tactile allodynia [253,254]. The increase in peripheral neuron sensitivity induced by tissue damage may be potentially attenuated by agents acting on α1-AR or α2-AR. 

#### 3.7.1. Clonidine

Clonidine, an α2-AR agonist, is an extremely potent antinociceptive agent, utilized systemically for chronic and acute pain treatment. Topical administration of clonidine elicits an antinociceptive effect, either in preclinical studies [255] or in clinical studies in patients with PDN [256,257,258]. According to Cochrane analysis of two RCTs, the efficacy of 0.1% topical clonidine has a medium level of evidence. However, topical clonidine has an excellent safety profile without central side effects observed following systemic administration [259]. Topical clonidine may provide some benefit in patients with PDN, and NNT for an additional beneficial outcome (NNTB) is 8.33, 95% CI 4.3 to 50 [259].

Clonidine is also an imidazoline receptor agonist and acts on these receptors located on peripheral nerve endings. The activation of imidazoline 2 receptors may possibly contribute to additional analgesic mechanisms of topically applied clonidine [260]. In preclinical studies, topical application of clonidine exerts a potent anti-inflammatory effect, which may be partially mediated by α2-AR and PGE inhibition. The anti-inflammatory properties may probably partially contribute to the analgesic effect of topical clonidine observed in LNP [261].

#### 3.7.2. Prazosin

Prazosin is an antagonist of α1-AR. Topically administered prazosin has been studied in healthy volunteers and patients with CRPS to date [262]. Prazosin 1% cream inhibited dynamic allodynia and punctate hyperalgesia in CRPS patients and adrenergic axon reflex vasodilatation in healthy volunteers. The potential target of prazosin may be the up-regulated α1-AR present on epidermal neurons, keratinocytes, and immune cells under neuropathic conditions [247,248].

#### 3.7.3. Other Drugs

Other drugs with a possible antinociceptive effect related to α1-AR blockade are antidepressants such as nortriptyline, imipramine, maprotiline, and milnacipran. Their antinociceptive effect via AR blockade has been observed after systemic administration in an animal formalin pain model [263]. Amitriptyline down-regulates the cortical and cerebellar α1-AR in the rat brain upon chronic treatment, but there are no data on peripheral regulation of α1-AR induced by systemic or topical amitriptyline in animal models and in humans [114].

### 3.8. Treatments Acting on SNAP-25 and 23

SNAP-25 is a component of the SNARE protein complex, which upon synaptic transmission is responsible for exocytotic neurotransmitter release. Through the assembly with syntaxin-1 and synaptobrevin, SNAP-25 mediates synaptic vesicle apposition to the presynaptic membrane, permitting their Ca^2+^-triggered fusion [264]. SNAP-23 is the ubiquitously expressed homologue of the neuronal SNAP-25, which is involved in synaptic vesicle fusion. Recently, it was shown that SNAP-23 mediates exocytosis in mast and epithelial cells and is involved in receptor trafficking [265]. The role of SNAP-25 in nociception in humans is confirmed, as specific gene polymorphisms for SNAP-25 are linked to chronic pain conditions, including NP and fibromyalgia [266].

#### Botulinum Toxin A

SNAP-25 and 23 are targeted by local administration of BTX-A. BTX-A is a potent toxin, which cleaves SNAP-25 and 23 and subsequently inhibits local release of neuropeptides and neurotransmitters involved in nociception, including SP [267], CGRP [268], and glutamate [269]. Moreover, our experimental results showed that BTX-A administered within the nerve terminals diminished nerve injury-evoked neuroimmune changes, primarily in the DRG and subsequently at the spinal cord level [270]. We observed as well that in DRG, the protein level of pronociceptive cytokines (IL-1β, IL-18) decreased and antinociceptive cytokines (IL-10, IL-1RA) increased following BTX-A peripheral application in an NP model [271]. Additionally, it was shown that an intra-nerve injection of BTX-A can stimulate the regeneration of an injured peripheral nerve and regrowing myelinated axons and improve the muscular reinnervation, so it generally speeds up sensorimotor recovery by stimulating myelinated axonal regeneration [272]. Animal studies showed that topical BTX-A administration through subcutaneous injection is followed by retrograde transport and transcytosis. This mechanism is possibly responsible for modulation of central sensitization and antinociception [273,274,275]. Several RCTs in humans showed effectiveness of BTX-A injections in patients with PHN, PDN, trigeminal neuralgia, and intractable neuropathic pain, such as poststroke pain and pain after spinal cord injury [276,277]. In clinical practice, local injections of BTX-A are recommended as the second-line treatment in patients with LNP [5,6]. BTX-A deserves special attention because when given topically through subcutaneous/intradermal injection, it can directly modulate both central and peripheral sensitizations [273,274,275].

### 3.9. Treatments Acting on Peripheral Opioid Receptors

The opioid receptors (OR)—μ (MOR), κ (KOR), δ (DOR), and nociceptin (NOR)—are GPCR and widely distributed in the CNS, peripheral neurons, and neuroendocrine (pituitary, adrenal), immune, and ectodermal cells. OR expressed on peripheral neurons and immune cells play a critical role in nociception and inflammation [278]. In preclinical studies, inflammation has been shown to increase mRNA transcription of OR in DRG, followed by enhanced axonal transport of de novo synthesized OR and increase in OR density in the peripheral nerve endings. Moreover, the lower pH of inflamed tissue may increase opioid–OR interactions and intracellular signaling, thus enhancing the analgesic effect of peripherally administered opioids [278]. Additionally, chemokines, cytokines, and other factors from inflamed tissue stimulate opioid peptide-containing immune cells to migrate to the site of injury. Immune cells (i.e., lymphocytes, granulocytes, monocytes, macrophages) in humans, rhesus monkeys, rats, and mice express not only opioid peptides but all types of OR—MOR, DOR, KOR, and NOR. Activation of MOR, DOR, and KOR on leukocytes stimulates release of opioid peptides, which bind to OR on peripheral sensory nerves and induce analgesia [278,279]. In an animal model of peripheral nerve injury, decreased MOR, but not DOR, expression in peripheral nerves and DRG has been observed. Nerve injury stimulates recruitment of immune cells at the injured site. Peripherally administered opioids in the case of decreased MOR expression in damaged neurons can act through the OR on immune cells (e.g., macrophages), leading indirectly to antinociception [279,280,281]. Moreover, the studies of human skin confirmed that epidermal keratinocytes express a functionally active OR. Human keratinocytes can both produce and bind β-endorphins, which further implies direct communication between peripheral nerve endings and skin cells, and their role in antinociception [282]. In dermatological diseases such as psoriasis, atopic dermatitis, or chronic wounds, dysregulation in the skin of both OR and their corresponding endogenous ligands has been observed. However, there are no data on dysregulation of skin OR under neuropathic conditions [283]. Data from preclinical studies suggest that peripheral OR on neurons, keratinocytes, and immune cells may serve as a target for opioids applied topically. Topical application may therefore be an alternative route of administration to systemic opioid treatment in patients with LNP and other localized pain syndromes, with less risk of side effects.

In patients with NP due to pachyonychia congenita, overexpression of NOR on epidermal keratinocytes and epidermal and dermal nerve fibers has been demonstrated [284]. Recently, NOR has been identified on human blood granulocytes as well [285]. Thus, NOR may be a promising target to manage NP. However, there are no available data on topical agents targeting NOR and utilized in clinical practice [278].

The main mechanism of opioid-induced analgesia is related to agonism to OR, localized in the presynaptic and postsynaptic neuronal membranes. Presynaptically, opioids inhibit neurotransmitter release by reducing Ca^2+^ influx. Postsynaptically, opioids open K^+^ channels, which hyperpolarize cell membranes and decrease the synaptic transmission. Opioids also inhibit adenylate cyclase, the enzyme converting ATP to cAMP [286]. Additionally, opioids such as morphine, tramadol, fentanyl, sufentanil, buprenorphine, and loperamide may influence Nav [138,139,140], inhibiting action potentials, which has been revealed in frog sciatic nerve models [138], and on unmyelinated mouse C-fibers [139].

#### 3.9.1. Loperamide

Loperamide is an opioid with higher binding affinity to MOR than to DOR or KOR [287]. Loperamide blocks Nav 1.7, Nav1.8, and Nav1.9 as well, and this inhibition may be the second mechanism of loperamide for pain relief beyond MOR agonism [140]. The bioavailability of loperamide after oral intake is only 0.3% and loperamide is extruded from the CNS actively; therefore, it does not act meaningfully at the CNS level [288]. The analgesic and anti-hyperalgesic effect of topically administered loperamide at different concentrations (0.5–5%) has been confirmed in NP models [289,290,291] and inflammatory pain models [292,293]. In preclinical studies, synergy between topical loperamide (MOR agonist) and oxymorphindole (DOR agonist) in reducing inflammatory hyperalgesia has been observed and was attributed to indirect inhibition of Nav1.8 [293,294]. In clinical settings, topical 5% loperamide has shown a beneficial effect in a patient with CIAP [295]. However, data on topical loperamide in clinical practice are limited to a single case report only.

#### 3.9.2. Morphine

The evidence on the effectiveness of topically administered morphine (agonist of MOR, DOR, KOR) comes mainly from clinical studies and case reports on cancer-related pain, i.e., cutaneous or mucosal lesions associated with local inflammation [296]. In one RCT, topical morphine (0.2% hydrogel or 0.2% ointment) showed a beneficial analgesic effect in a cancer patient with pain due to mucosal lesions and skin ulcers [297]. Data on topical use of morphine and other opioids in patients with LNP are lacking.

#### 3.9.3. Other Drugs Modulating Opioid System

The endogenous opioid system may be indirectly modulated by drugs such as antidepressants [113], ketamine [204,205], diclofenac [231], and CBD [298], but whether these mechanisms contribute to the analgesic effect upon topical application in humans remains unclear. 

### 3.10. Treatments Acting on Peripheral Cannabinoid Receptors

Numerous studies indicate a modulatory effect of the endocannabinoid system in NP [299]. In the periphery, CB1 receptors are expressed on nociceptive peripheral nerve endings and DRG, whereas CB2 receptors are located mainly on immune cells and keratinocytes [300,301]. Preclinical data suggest that agents acting on CB1 receptors may evoke a beneficial effect against NP, and those acting on CB2 may evoke a beneficial effect against inflammatory pain [300,301]. Either CB1 or CB2 receptors may be targeted by cannabinoids administered topically, influencing the activity of both neuronal and non-neuronal cells. Preclinical studies reveal that activation of CB2 receptors on keratinocytes stimulates them to release β-endorphins, which in turn act at local neuronal MOR, inhibiting nociception [298]. CB1 receptor agonists attenuate mast cell activation and subsequent local inflammation in a model of dermatitis [302]. Theoretically, substances acting on peripheral CB receptors may provide an analgesic effect in patients with LNP, as these target cells involved in peripheral sensitization.

#### Cannabidiol

CBD is one of the main phyto-cannabinoids found in Cannabis sativa and indica. CBD is a lipophilic, multi-target drug, whose central antianxiety/antipsychotic effect is probably related to interaction with CB1 receptors in the CNS [303]. However, whether the antinociceptive effect of CBD is attributed to interactions with CB1 is still discussed [303]. An NP mouse model caused by paclitaxel showed that intraperitoneal administration of CBD reduced allodynia and NP behavior through the influence of serotonin receptors [304]. In preclinical studies, CBD directly interacted with Nav and Kv channels, whose combined effects are reduction in channel hyperexcitability [143]. Moreover, CBD may diminish neuronal hyperexcitability through other mechanisms of action: inhibition of GPCR (GPR55) at excitatory synapses, desensitization of TRPV1, and modulation of the adenosine system [148,150,151,305]. These mechanisms may be responsible for the antiepileptic properties of CBD and, together with MOR and DOR modulation, are suggested to play a role in antinociception induced by CBD [303]. Whether and to what extent these mechanisms are responsible for analgesia upon topical application in humans are unclear.

Clinical observations indicate that topical administration of CBD, mixed with other anti-inflammatory phyto-derived products, in oil, gel, cream, or spray, may exert beneficial analgesic, immunosuppressive, and anti-inflammatory effects in humans, as reviewed recently [306]. In an RCT testing topical CBD cream (250 mg CBD/3 fl. oz, around 8.3%) in 29 patients with LNP compared to placebo, CBD oil showed a more pronounced pain-reducing effect than placebo [144]. The preclinical data and clinical observations support the idea of topical administration of CBD in pain states. However, the data on analgesic efficacy in LNP conditions are limited to a single RCT [144].

## 4. Topical Treatments in Patients with Neuropathic Pain—Summary of Possible Mechanisms of Antinociception and Future Directions

Research conducted during the last decade has identified many potential peripheral mechanisms for NP, pointing at peripheral sensitization as the target for therapeutic strategies in patients with NP [11]. Peripheral nerve endings express a variety of excitatory and inhibitory receptors, ion channels, and proteins, such as Nav, NMDAR, VGCCs, α-AR, TRPV1, TRPM8, SNAP-25 and 23, GABAR, COX-2, OR, and CB. Some of them are also expressed by non-neuronal cells (Figure 1). Moreover, peripheral neurons interact closely via several signaling molecules with immunocompetent cells and keratinocytes and this interplay under pathological conditions is responsible for neuronal hyperexcitability and NP generation (Figure 1) [10,11]. These facts enable the development of pharmacotherapies specifically targeting peripheral mechanisms, e.g., topical analgesics. Preclinical data discussed in this paper showed that active molecules applied in topical formulations in humans (lidocaine, capsaicin, BTX-A, clonidine, doxepin, phenytoin, amitriptyline, ketamine, CBD, funapide, baclofen, ambroxol, gabapentin, prazosin, menthol, diclofenac, and loperamide) exert multiple mechanisms of action and can directly and/or indirectly modulate distinct molecular/cellular targets and pathways in the nociceptive system. Thus, active molecules, released from a topical formulation and targeting the elements of the nociceptive pathway, may possibly induce an analgesic effect. However, to which extent each of the mechanisms and molecular/cellular targets contributes to the analgesic effect observed in humans is not clear. Clinical data suggest that more selective agents, such as funapide, induce less pronounced analgesia in comparison with non-selective drugs such as lidocaine [5,6,46]. In preclinical studies, lidocaine at different concentrations and routes of administration was shown to act on several ion channels and receptors involved in nociception [48], while funapide acts on Nav1.7 only [46]. However, the effectiveness of multitargeted topical drugs and/or combinations of drugs of different mechanisms of action needs further evaluation in “head-to-head” clinical trials. Table 1 summarizes the preclinical data on possible molecular/cellular mechanisms involved in the antinociceptive effect of topical treatments utilized in clinical practice.

Many topical agents are used for the treatment of NP in humans (Table 1), despite data coming from case reports, observational studies or single RCTs. The ongoing clinical trials in NP are aimed at comparison between systemic and topical analgesics and, what is more important, at the effectiveness of topical analgesics in special patient populations (i.e., cancer patients). The future directions in pain medicine will also focus on the assessment of optimal concentrations of topical formulations. Thus far, nearly no dose-finding trials for topical analgesics have been conducted in clinical practice. Therefore, the optimal concentration and concentration-related mechanism of action are unknown for topical analgesics and these also need further preclinical and clinical assessment. Table 2 summarizes the data on currently ongoing and recruiting clinical trials with topical treatments in patients with LNP. The data are based on information available on websites: ClinicalTrials.gov provided by the U.S. National Library of Medicine, and the EU Clinical Trials Register [307,308]. The search was conducted with the terms “topical neuropathic” and “peripheral neuropathic”.

## 5. Conclusions

Peripheral nerve injury induces functional and structural changes in neuronal and non-neuronal cells, which release numerous signaling molecules in response to the damage. In turn, these mediators modulate corresponding receptors on cell membranes, creating vicious circles of interactions. These maladaptive mechanisms taken together contribute to the sensitization of peripheral nerve endings and enhanced peripheral input leading to neuropathic pain. At present, topical lidocaine, capsaicin, BTX-A, clonidine, doxepin, phenytoin, amitriptyline, ketamine, CBD, funapide, baclofen, ambroxol, gabapentin, prazosin, menthol, diclofenac, and loperamide are being used in a variety of LNP states in humans, bringing pain relief. To date, the evidence from several RCTs and reviews supports 5% lidocaine patches, 8% capsaicin patches, and BTX-A injections as effective treatments in patients with LNP. In turn, single RCTs support evidence of doxepin, funapide, diclofenac, baclofen, clonidine, loperamide, and CBD in LNP. Topical administration of phenytoin, ambroxol, and prazosin is supported by observational clinical studies only. For topical amitriptyline, menthol, and gabapentin, evidence comes from case reports and case series. For topical ketamine and baclofen, data supporting their effectiveness are provided by both single RCTs and case series. The possible mechanisms of antinociception of topical treatments are discussed in this paper. However, which mechanism and to what extent it contributes to pain relief observed in humans are still unclear. In patients suffering from LNP, multiple mechanisms are involved in pain generation; however, in clinical practice, simple tools assessing underlying pain mechanisms are still lacking. On the other hand, directly testing topical analgesics in a single- or double-blind placebo-controlled manner, in line with personalized medicine, can give a rapid clinical answer concerning pain reduction in subjects with LNP. This review may help clinicians in making decisions regarding whether and which topical treatment may be a beneficial treatment option, and what clinical effect patients suffering from LNP may expect.

## Figures and Tables

**Figure 1 pharmaceutics-13-00450-f001:**
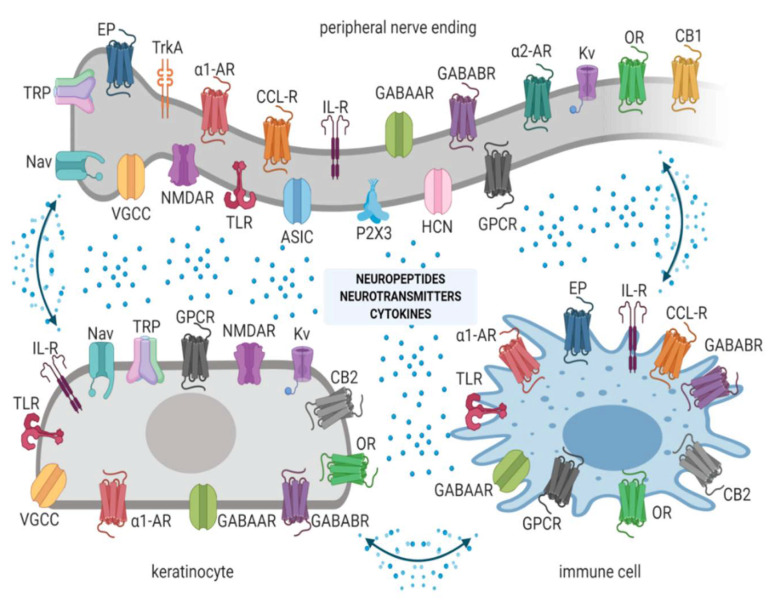
Peripheral nerve endings, keratinocytes, and immune cells express ion channels and receptors and release numerous signaling molecules to create a complex interaction, involved in physiological nociception and neuropathic pain (NP) generation. Abbreviations: Nav—voltage-gated sodium channels, TRP—transient receptor potential channels, VGCCs—voltage-gated calcium channels, NMDAR—N-methyl-D-aspartate receptors, ASIC—acid-sensing ion channels, TLR—Toll-like receptors, α1-AR—α1 adreno receptors, α2-AR—α2 adreno receptors, EP—prostaglandin E2 receptors, GABAAR—gamma-aminobutyric acid receptors A, GABABR—gamma-aminobutyric acid receptors B, Kv—voltage-gated potassium channels, OR—opioid receptors, CB1, CB2—cannabinoid receptors type 1 or 2, CCL-R—chemokine receptors, IL-R—interleukin receptors, TrkA—tropomyosin receptor kinase A, HCN—hyperpolarization-activated cyclic nucleotide-gated channels, P2X3—P2X purinoceptors 3, GPCR—G protein-coupled receptors, TLR—Toll-like receptors. Created with BioRender.com.

**Figure 2 pharmaceutics-13-00450-f002:**
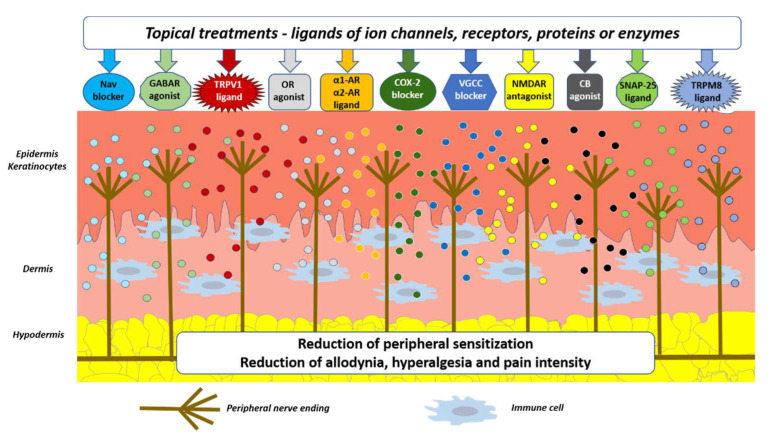
Topical treatments utilized in clinical practice and their molecular/cellular mechanisms in patients suffering from localized NP (LNP). Active molecules from topically applied drug formulations modulate the corresponding ion channels, receptors, enzymes, or proteins on neuronal and non-neuronal cells. Abbreviations: Nav—voltage-gated sodium channels, TRPV1—transient receptor potential vanilloid 1, VGCCs—voltage-gated calcium channels, NMDAR—N-methyl-D-aspartate receptors, α1-AR—α1 adreno receptors, α2-AR—α2 adreno receptors, GABAR—gamma-aminobutyric acid receptors, CB—cannabinoid receptors, COX-2—cyclooxygenase 2, SNAP-25—synaptosome associated protein 25, OR—opioid receptors, TRPM8—transient receptor potential melastatin 8.

**Table 1 pharmaceutics-13-00450-t001:** Possible direct or indirect mechanisms of antinociceptive action of topical agents, cells influenced by a given agent, and form of drug used in clinical trials and/or daily practice in patients with LNP. Presented references refer to possible mechanisms of action. Abbreviations: Nav—voltage-gated sodium channels, TRPV1—transient receptor potential vanilloid 1, TRPA1—transient receptor potential ankyrin 1, TRPM8—transient receptor potential melastatin 8, TRPM3—transient receptor potential melastatin 3, mAChR—muscarinic acetylcholine receptors, nAChR—nicotinic acetylcholine receptors, VGCCs—voltage-gated calcium channels, L-VGCCs—L-type voltage-gated calcium channels, NMDAR—N-methyl-d-aspartate receptors, ASIC—acid-sensing ion channels, P2X7—P2X purinoceptor 7, PGE2—prostaglandin E2, GABAAR —gamma-aminobutyric acid receptors A, GABABR—gamma-aminobutyric acid receptors B, Kv—voltage-gated potassium channels, K+—potassium, OR—opioid receptors, CB1—cannabinoid receptor type 1, 5-HT—serotonin, 5-HT-R—serotonin receptors, GPCR—G protein-coupled receptors, TLR4—Toll-like receptor 4, HCN—hyperpolarization-activated cyclic nucleotide-gated channels, NGF—nerve growth factor, TrkA—tropomyosin receptor kinase A, Gly—glycine, α1-AR—α1 adreno receptor, α2-AR—α2 adreno receptor, H-R—histamine receptor, DA—dopamine, NA—noradrenaline, NO—nitrous oxide, COX-2—cyclooxygenase 2, OR—opioid receptors, KOR—κ opioid receptors, SNAP—synaptosome associated proteins, I2-R—imidazoline receptors, EMLA—eutectic mixture of local anesthetics.

Topical Agent	Direct or Indirect Mechanism of Action	Cellular Targets	Reference	Form of Drug
**Lidocaine**	Nav blockade mAChR blockade TRPA1 desensitization NMDAR antagonism ASIC blockade HCN blockade TLR4 inhibition Kv blockade VGCC blockade P2X7 inhibition NGF/TrkA modulation Gly system modulation Anti-inflammatory effect	Neurons Keratinocytes Immune cells Schwann cells	[39,45,54,55,56,57,58,59,60,61,62,63,64,65,66,67,68,69,70,71,72,73,74,75,76,77]	5% patch EMLA cream EMLA patch 2–11% cream, gel 10% spray 7% cream combined with 7% tetracaine
**Phenytoin**	Nav blockade L-VGCC blockade GABAAR modulation Anti-inflammatory effect	Neurons Keratinocytes Immune cells	[40,85,86,87,88,89]	5–30% cream
**Ambroxol**	Nav blockade Anti-inflammatory effect	Neurons Keratinocytes Immune cells	[41,42,103,104]	20% cream
**Antidepressants:** **-Amitriptyline** **-Doxepin**	Nav blockade VGCC blockade NMDAR antagonism Kv activation α1-AR down-regulation TRPA1 desensitization GABABR modulation 5-HT-R blockade H-R blockade mAChR blockade Reduction in NO, PGE2 NA, 5-HT, DA and adenosine reuptake inhibition Opioid system modulation	Neurons Keratinocytes	[43,44,107,108,109,110,111,112,113,114,115,116,117,118,263]	Amitriptyline: 1–10% cream Doxepin: 3.3%, 5% cream
**Funapide**	Nav1.7 blockade	Neurons Keratinocytes	[46]	hydrogel
**Capsaicin**	TRPV1 activation	Neurons Keratinocytes	[147]	8% patch 0.025–0.1% cream
**Menthol**	TRPM8 activation GABAAR activation VGCC blockade TRPA1 inhibition nAChR blockade	Neurons Keratinocytes Immune cells	[142,171,172,173,174,175]	2.5–16% gel 7.5%, 16% cream 16% solution 16% spray 1.25–16% patch
**Gabapentin**	VGCC blockade NMDA blockade Kv activation Anti-inflammatory effect	Neurons Keratinocytes	[186,187,188,189]	2–6% cream
**Ketamine**	NMDA antagonism Nav blockade OR re-sensitization L-VGCC inhibition GABAA activation mAchR inhibition nAChR inhibition TLR4 inhibition Nav blockade	Neurons Keratinocytes Immune cells Schwann cells	[141,204,205,206,207,208,209,210,211]	0.5–20% cream
**Diclofenac**	COX-2 inhibition NMDAR antagonism TRPV1, TRPA1, TRPM3 ligand Nav blockade VGCC inhibition K^+^ channels modulation α1-AR interaction KOR modulation	Immune cells Neurons Keratinocytes Schwann cells	[138,169,226,228,229,232]	1–1.5% gel 140 mg patch
**Baclofen**	GABABR agonism	Neurons Keratinocytes Immune cells Schwann cells	[239]	2%, 5% cream
**Clonidine**	α2-AR activation Nav blockade I2-R agonism Anti-inflammatory effect	Neurons Keratinocytes Immune cells	[138] [255,260,261]	0.1%, 0.2% gel 0.1%, 0.2% cream
**Prazosin**	α1-AR blockade	Neurons Keratinocytes Immune cells	[262]	1% cream
**BTX-A**	SNAP-25 SNAP-23 Anti-inflammatory effect	Neurons Immune cells Microglia Astroglia	[267,268,269,271]	Intradermal injections
**Loperamide**	OR agonism Nav blockade	Neurons Immune cells Keratinocytes	[138,139,140,286]	5% loperamide cream
**Cannabidiol**	CB1 interaction Nav and Kv interaction TRPV1 desensitization 5-HT-R modulation Adenosine system modulation Opioid system modulation Synaptic GPCR interaction Anti-inflammatory effect	Neurons Immune cells Keratinocytes	[143,148,150,151,303,304,305]	ointment, cream cream: 250 mg CBD/3 fl. (around 8.3%)

**Table 2 pharmaceutics-13-00450-t002:** Clinical trials currently ongoing and recruiting patients with peripheral neuropathic pain syndromes of different origin. Abbreviations: RCT—randomized controlled trial, vs. – versus, BTX-A—botulinum toxin type A.

Title of the Study	Formulations/Drugs Studied	Type of Study
The Effects of Topical Treatment with Clonidine + Pentoxifylline in Patients with Neuropathic Pain	Solution of clonidine (0.1%) + pentoxifylline (5%) vs. Placebo	RCT Triple blind
Multicentric, Open, Randomized Study Comparing Topical Treatment by Patch of Capsaicin to 8% (Qutenza) to Pregabalin Oral in the Early Treatment of Neuropathic Pain After Primary Surgery for Breast Cancer	Capsaicin 8% patch vs. Oral pregabalin	RCT Open label
A Phase II RCT of Topical Menthol Gel vs. Placebo in the Treatment of Chemotherapy Induced Peripheral Neuropathic Pain	Menthol gel vs. Placebo	RCT Triple blind
Clinical Trial Assessing the Efficacy of Capsaicin Patch (Qutenza®) in Cancer Patients with Neuropathic Pain	Capsaicin 8% patch	Open-label clinical trial
Intraoral Administration of Onabotulinum Toxin A for Continuous Neuropathic Pain: a Single Subject Experimental Design	BTX-A	Open-label clinical trial
A Multicentre, Single-Arm, Open-Label Study of the Repeated Administration of QUTENZA for the Treatment of Peripheral Neuropathic Pain	Capsaicin 8% patch	Open-label clinical trial
Is there a correlation between the pain relief and the A-delta- and C-fiber function after topical application of lidocaine (5%) in patients with peripheral neuropathic pain?	Lidocaine 5% patch vs. Placebo	RCT Double blind
Amitriptyline 10% and ketamine 10% cream in neuropathic pain: A randomised, double-blind, placebo-controlled cross-over pilot study with a three months open follow-up	Amitriptyline 10% cream vs. Ketamine 10% cream vs. Placebo	RCT Double blind
Enrichment randomized double-blind, placebo-controlled cross-over trial with PHEnytoin cream in patients with painful chronic idiopathic axonal polyNEuropathy	phenytoin 10% cream vs. Phenytoin 20% cream vs. Placebo	RCT Double blind

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
