# Peer review of "Topical Treatments and Their Molecular/Cellular Mechanisms in Patients with Peripheral Neuropathic Pain—Narrative Review"

_pharmaceutics, 2021, doi:10.3390/pharmaceutics13040450_

Round 1

Reviewer 1 Report

The authors reported a literature review comprising the main topical treatments for peripheral sensitization and pain. The review is well documented and organized and it gathered important information on various types of drugs used for topical application and its mechanism of action. However, there are still editorial issues to fix and rearrange the manuscript. Please find below some comments/suggestions which might improve the quality of the manuscript:

  1. Figures 1and 2-copyrights permissions were not mentioned; the authors are requested to ask for the copyright permissions and to mention them.
  2. The authors presented the description of the drugs’ mechanism by inserting for each a sub-subtitle within section 3.1; however, the other sections are not numerated in the same manner. Therefore, the authors are requested to uniformize the numeration style throughout the manuscript.
  3. The authors mentioned for each drug some concentrations; the authors did not include any information on the maximum/optimum concentration admitted and the influence of the concentration on the mechanism.
  4. The authors included within the conclusion part a table comprising information on various types of drugs together with mechanism and formulation. This type of summarized information could be more useful within the text to support some of the details. The authors should re-organize the conclusion part so that an outcome be clear.

Reviewer 2 Report

The manuscript showed a narrative review of studies related with the molecular/cellular mechanism of drugs available in topical formulations utilized in clinical practice and their effectiveness in clinical studies in patients with peripheral neuropathic pain. However, the review suffers from several incompetencies which address the given issues:

Page 1; Line 27: Abstract- What methods authors applied to review? Any customization on selection of the available piece of evidences.

Page 1; Line 29-33: The results of the narrative review in terms of author's critical review should reflect in the abstract.

Page 2: Line 70: Required to add a section as Methods or data search immediate after introduction which will justify the construction of the manuscript in a critical way. It should include the selection criteria such as the use of what kind of data base sources used (Such as Scopus, MEDLINE, PubMed, Cochrane and ScienceDirect etc.), how did the filter occur (how many years of studies, and types of study including in vitro, in vivo and clinical studies). Further, the clinical significance of the studies in each subsection missing.

Page 4; Line 130: Need to brief selective clinical studies in the related conditions.

Section descriptions: Each section described very well results of the past studies however, author’s evaluation and conclusions significant outcome with future directions were missing?

Reviewer 3 Report

The manuscript consists of a well-designed review on the topical use of drugs to treat neuropathic pain, particularly their mechanism of action. There are not many recent reviews focusing the mechanisms. As the subject is becoming too broad, this review is an interesting contribution because of its focus. It fits the journal’s scope.

Authors should consider the following comments:

  1. Line 130. … “data show” instead of “data shows”.
  2. Line 909 … “there are no data” instead of “there is no data”.
  3. Topical application of morphine has been extensively studied, particularly in painful cancer wounds. This aspect should be better addressed in the paper.
  4. Clinical trials are frequently mentioned, but the paper would benefit from a table indicating the main clinical trials currently ongoing with drugs intended for local treatment of NP.
  5. The English language and style is a bit repetitive and could be improved.

Round 2

Reviewer 2 Report

The manuscript has been improved from the previous version and addressed all the comments.